# Superficial Venous Thrombosis: A Comprehensive Review

**DOI:** 10.3390/healthcare12040500

**Published:** 2024-02-19

**Authors:** Marco Mangiafico, Luca Costanzo

**Affiliations:** 1Unit of Internal Medicine, Policlinico “G. Rodolico-San Marco” University Hospital, University of Catania, 95123 Catania, Italy; marcomangiafico@hotmail.it; 2Unit of Angiology, Department of Cardio-Thoraco-Vascular, Policlinico “G. Rodolico-San Marco” University Hospital, University of Catania, 95123 Catania, Italy

**Keywords:** superficial venous thrombosis, varicose vein, venous thromboembolism, epidemiology

## Abstract

Superficial venous thrombosis (SVT), an inflammatory–thrombotic process of a superficial vein, is a relatively common event that may have several different underlying causes. This phenomenon has been generally considered benign, and its prevalence has been historically underestimated; the estimated incidence ranges from about 0.3 to 1.5 event per 1000 person-years, while the prevalence is approximately 3 to 11%, with different reports depending on the population studied. However, such pathology is not free of complications; indeed, it could extend to the deep circulation and embolize to pulmonary circulation. For this reason, an ultrasound examination is recommended to evaluate the extension of SVT and to exclude the involvement of deep circulation. Also, SVT may be costly, especially in the case of recurrence. Therefore, accurate management is necessary to prevent sequelae and costs related to the disease. This review aims to analyse the epidemiology of SVT, its complications, optimal medical treatment, and open questions with future perspectives.

## 1. Introduction

Superficial venous thrombosis (SVT), often also called superficial thrombophlebitis, is defined as a pathologic process that involves a vein of the superficial circulation, with inflammation and thrombosis of the vessel [1,2].

Venous disease has been known since ancient times. The first documentation of the disease is from the “Ebers papyrus”, dated around 1500 Before Christ (BC) in which a “tortuous and solid with many knots” vein was described; however, no indication for treatment was provided [3]. A varicose vein was depicted on a votive tablet dated 400 BC and found at the base of the Acropolis [4]. The first case of thrombosis reported in the scientific literature was described by Guillaume De Saint Pathus, a Franciscan monk, who reported symptoms characteristic of deep venous thrombosis (DVT) in a young Norman cobbler, such as worsening oedema and pain in the right calf spreading to the thigh [5].

The first description of the thrombotic involvement of a superficial vessel was attributed to Armand Trousseau, who noted the association of superficial migratory thrombophlebitis and visceral malignancy in 1865 [6].

SVT has been regarded as a relatively benign and self-limiting disease; however, troubling cases are possible, such as DVT or pulmonary embolism (PE). The objective of this review is to analyse the epidemiology of SVT, its complications and their outcomes, and the optimal medical therapy, with a focus on answering open questions about the best management.

## 2. Epidemiology

SVT is a relatively common disease that mostly affects the lower limbs. The incidence is estimated to be 0.3–0.6 events per 1000 person-years in young people and 0.7–1.5 events per 1000 person-years in older patients [7,8], not so different than DVT, which is estimated to be about 1 in every 1000 cases [9,10]. However, the real incidence of STV is probably underestimated.

The STEPH study was a descriptive, multicentre, community-based study conducted over 1 year in the adult resident population of an urban area in France. The study included 265,687 adults and 171 of them had symptomatic SVT, confirmed with ultrasound performed by vascular specialists. The measured annual diagnosis rate of SVT was 0.64% of adults [95% confidence interval (CI) 0.55% to 0.74%]; however, it is possible that the primary care setting underestimated the true prevalence [7].

Another retrospective cohort study evaluated the Dutch population and the annual incidence of SVT by using healthcare-coded data from the Utrecht General Practitioner Network database. The diagnosis of SVT was clinical, through the description of typical signs and symptoms. The incidence of SVT events recorded in primary care was 1.31 events per 1000 person-years of follow-up [11].

The Tecumseh Community Health Study (TCHS), a longitudinal clinical study conducted in the United States starting in 1957, estimated that there were 123,000 cases of SVT per year [8].

Another study analysed the prevalence of clinically diagnosed SVT in the Swiss population. The prevalence reported was about 3–11% depending on the population analysed [12]. Notably, the latter two studies are old and based only on clinical diagnoses; therefore, the real prevalence could be underestimated [13].

The venous thrombosis prevalence rate was found to be 2.9% in women and 0.8% in men aged ≥49 in an Italian study (Anemone) that used a self-managed questionnaire to investigate the prevalence of the condition in blood donors. After testing for confounding potentials, a significant and independent association was found between a history of venous thrombosis and age [Odds ratio (OR): 1.03, 95% CI 1.01–1.05], varicose veins (OR: 15.8, 95% CI 7.7–32.6), gypsum/bed rest (OR: 2.3, 95% CI 1.0–5.3), and transfusion (OR: 5.1, 95% CI 1.3–19.5). The study revealed that low-risk individuals share the same risk factors for SVT as those in secondary care [14].

Another interesting point is the seasonality of SVT. A small retrospective study showed an increase in the summer months; a possible explanation could be the poor compliance of patients to pharmacologic therapy and elastic stockings [15]. Other reports [16,17] did not confirm such an observation, although a small increase was observed in summer [17].

In conclusion, it is worth noting that most data about patients with SVT are derived from old studies with methodological limitations. New studies using diagnostic technologies such as ultrasound could allow for a better epidemiological definition of pathology and complications [18].

## 3. Risk Factors for SVT

The risk factors for SVT and DVT are similar: advanced age, varicose veins, pregnancy, post-operative states, immobilization, malignant neoplasms, autoimmune diseases, obesity, trauma, hypercoagulable states, use of oral contraceptives or hormonal therapies, previous episodes of DVT/PE, vascular access, infusion of hypertonic solutions or endothelial damaging substances, and autoimmune diseases [19]. Unlike DVT, varicose veins are the primary risk factor for lower-limb SVT, and they are found in 90% of cases [20]. Among autoimmune disorders, Behcet’s disease has been associated with SVT onset. Also, patients with Buerger’s disease are particularly susceptible to SVT. Interestingly, in such cases a peculiar inflammation of the three layers of the vessel has been described [21].

In DVT development, Virchow’s triad consisting of altered blood flow, changes in the vessel walls, and abnormal coagulation are recognized to be possible risk factors. While stasis and the trauma of the endothelium have been cited as causes of SVT, the importance of hypercoagulability has been questioned [22].

Karathanos and colleagues investigated the impact of specific risk factors on the impact of varicose veins and SVT in a cohort of 128 patients with varicose veins and SVT, diagnosed through ultrasonography compression. Age, male sex, obesity, and thrombophilia defects such as Protein S deficiency were significantly associated with SVT among patients with varicose veins and moderate disease [23].

Studies have indicated that people with SVT have a prevalence of various thrombophilic factors that is approximately 2–3 times higher than those without SVT [24,25]. Indeed, the common presence of hereditary thrombophilia, such as factor V Leiden, prothrombin mutation G20210A, or protein C, S, or antithrombin (AT) deficiency, indicated a pathogenesis similar to DVT [26]. A high prevalence of factor V Leiden was discovered in an Italian population with SVT in non-varicose veins [27]. In another Italian retrospective study of about 2000 patients, SVT was not found to be linked to G20210A prothrombin mutation, positivity for antiphospholipid antibodies, hyperhomocysteinemia, and high factor VIII levels; conversely, the association with SVT and protein C, S, and AT deficiency was reported [28].

Table 1 summarises the data.

The presence of cancer in SVT was assessed in a retrospective study that reported a prevalence of 8.7%. Also, the authors found a significant association among age, thrombophilia, male gender, non-varicose vein SVT, and cancer with concurrency of DVT/PE. Particularly, the presence of cancer was the strongest risk factor for DVT/PE [29].

SVT of the upper limbs and neck is mainly secondary to iatrogenic causes related to the use of intravenous catheters for infusion of chemotherapy drugs, parenteral nutrition, or other drugs and is a relatively frequent complication in hospitalized patients, usually in the forearm or hand [30].

Another specific and rare type of SVT is Mondor’s disease, which can affect the veins of the anterolateral chest wall, penis, and axillary veins, often after arm surgery [31].

## 4. Cost Related to SVT

SVT occurrence also impacts health care spending. In 2007, Cesarone and coworkers evaluated the average cost of an SVT event at about EUR 1180, which could reach up to EUR 5000 in the case of complex surgical cases [32]. Kim Y. and coworkers reviewed the cost related to chronic venous insufficiency, the condition most associated with SVT, in the United States. The authors estimated a total cost of treatment of around USD 3 billion per year among 25 million adults [33]. Also, the presence of low socioeconomic status, black race, and male gender predicted an advanced stage of chronic venous disease (CVD) classification at the initial presentation. Notably, the possible DVT complication or PE may increase costs, especially in the case of recurrence [34]. The findings suggest that improved mechanisms are needed to identify venous disease and at-risk patients before reaching advanced disease progression in known disadvantaged patient populations [35]. Considering the high frequency of the problem, any improvement in management contributes to an important reduction in medical costs.

## 5. Prevalence of DVT and/or PE at Diagnosis of SVT

In the past years, the clinical impact of SVT may have been underestimated, as it was considered a benign and self-limiting condition that required symptomatic treatment [36]. However, several reports described the occurrence of extension to the deep venous system and the embolization to pulmonary circulation [37].

In the STEPH study, among 171 patients with SVT, concomitant DVT and/or PE were reported in 24.6% and 4.7% of cases, respectively. All diagnoses were obtained with ultrasonographic compression performed by vascular specialists. Notably, such events occurred most frequently when the SVT involved the great saphenous vein (GSV) or was located at or above the knee and/or extended to the perforating veins [7].

A similar incidence of DVT and/or PE (24.9%) was found in the Prospective Observational Superficial Thrombophlebitis (POST) study [38] that included consecutive SVT more than 5 cm in length diagnosed via ultrasonography compression.

In the prospective multicentre OPTIMEV study, ultrasonography compression diagnosis revealed that 29% of patients with SVT also had a DVT. Particularly, a concurrent DVT was found in 39.4% of SVT in non-varicose veins [39].

Finally, in a meta-analysis by Di Minno and coworkers based on 4 358 patients in 22 studies, the concurrent prevalence of DVT and PE at diagnosis of SVT was 18.1% (95% CI: 13.9% and 23.3%) and 6.9% (95% CI: 3.9% and 11.8%), respectively [40]. The authors of the meta-analysis acknowledge the limitations of the data due to the heterogeneity and different methodologies of the studies included. The prevalence of DVT/PE is not negligible, but it is advisable to conduct prospective studies, especially in patients considered to be at high risk [40].

According to the studies examined [7,38,39], ultrasound compression, venography, computer tomography, scintigraphy, and, in some cases, autopsies have been used to diagnose DVT and PE. The source of this information is unclear, symptoms are not reported, and the severity scores of PE that have a significant impact on mortality and complications are not reported.

Table 2 summarises the data.

## 6. The Outcome of SVT in Pregnancy

Pregnancy-associated SVT is of particular interest as it is not free from complications. In a cohort study conducted in Denmark, the estimated prevalence of SVT in pregnancy was about 0.1%. The authors also assessed the timing of SVT onset from conception up to three months after delivery. SVT occurred mainly during the post-partum period with an incidence rate of 1.6 per 1000 person-years (95% CI 1.4–1.7); during pregnancy, the incidence slightly increased in the trimesters: 0.1 (95% CI 0.1–0.2) in the first trimester, 0.2 (95% CI 0.2–0.3) in the second trimester, and 0.5 (95% CI 0.5–0.6) in the third trimester. Notably, in 10.4% of women with previous SVT, VTE was diagnosed, underlining the importance of SVT in such a typology of patients [41].

## 7. Risk Factors for Thromboembolic Complications

The most serious complication associated with SVT is the extension to the deep vein and the embolization to pulmonary circulation. In the literature, several factors were found to be associated with an increased risk of such complications. In the POST observational study, among the 586 patients with isolated SVT who were followed up for three months, 10.2% developed thromboembolic complications: 0.5% PE, 2.8% DVT, 3.3% extension of SVT, and 1.9% recurrence of SVT. The multivariate analysis showed that male sex [Hazard Ratio (HR): 2.63, 95% CI 1.42–4.86, *p* = 0.002], a history of DVT or PE (HR: 2.18, 95% CI 1.15–4.12, *p* = 0.016), non-varicose vein (HR 2.06, 95% CI 1.01–4.25, *p* = 0.049), and previous cancer (HR: 3.12, 95% CI 1.15–8.47, *p* = 0.067) were independent risk factors for symptomatic thromboembolic events at 3 months, including recurrence or extension of SVT in patients with isolated SVT at inclusion [38].

In the MEGA study, a history of previous SVT was associated with a 4-fold (OR: 3.9, 95% CI 3.0–5.1) and 6-fold (OR: 6.3, 95% CI 5.0–8.0) increased risk of PE and DVT, respectively. According to the authors, excluding clinical diagnosis from analysis is incorrect because it does not accurately reflect the real world [42].

In a nationwide population-based registry conducted in Denmark during a period when SVT was not treated routinely with anticoagulants, 10 973 patients with a first-time diagnosis of SVT, mostly clinical, were identified between 1980 and 2012. The authors drew attention to the first three months after SVT as there is a 70-fold increased risk of VTE complication (adjusted HR: 71.40, 95% CI 60.16–84.74); however, a decreased but persistent fivefold increased risk remained during the next 5 years (adjusted HR: 5.05, 95% CI 4.61–5.54). Also in this registry, male sex was found to be more associated with thromboembolic complications such as DVT and PE [43].

Male sex (OR: 3.5, 95% CI 1.1–11.3) and inpatient status (OR: 4.5, 95% CI 1.3–15.3) were confirmed as predictors of VTE complication at three months in the case of isolated SVT in the OPTIMEV study. In this registry, SVT occurring on non-varicose veins (OR: 1.8, 95% CI 1.1–2.7, *p* < 0.005), age > 75 years (OR: 2.9, 95% CI 1.5–5.9, *p* < 0.005), active cancer (OR: 2.6, 95% CI 1.3–5.2, *p* < 0.005), and inpatient status (OR: 2.3, 95% CI 1.2–2.4, *p* < 0.005) were also independently associated with an increased risk of concurrent DVT at presentation [39].

Lastly, thrombophilic disorders such as FV Leiden, prothrombin G20210A mutation, protein C, protein S, and AT deficiency have been identified as factors for the progression of SVT to the deep venous system [44,45].

Despite several studies investigating SVT, there are no risk prediction models for predicting complications in SVT. A project is currently underway to analyse data from four large patient registries and develop and validate new models to evaluate three types of complications: the first is to assess the poor control of SVT symptoms and/or the possibility of SVT extending to the sapheno–femoral junction (SFJ) within 14 days of diagnosis; the second model aims to evaluate the progression of the clot towards the DVT and/or PE onset within 45 days after diagnosis of SVT; and for the third model, the endpoint is SVT recurrence within 12 months. The challenge is to develop decision models to identify patients with a higher risk of developing complications and to support therapeutic decisions [46]. Risk factors for thromboembolic complications are summarised in Table 3.

## 8. Diagnosis of SVT

The diagnosis of SVT is essentially clinical, as a superficial thrombosed venous vessel is usually manifested by a red, tender, and palpable cord of the skin [47]. Differential diagnosis can be clinically difficult with cellulite, panniculitis, erythema nodosus, insect bites, and lymphangitis [48].

Unlike DVT and PE, D-dimer is not a valid tool for the diagnosis of lower limb SVT [49]. The specificity and sensitivity in the diagnosis of SVT are not adequate for clinical use, as confirmed by some authors [50,51]. Sartori and coworkers analysed the role of D-dimer in the diagnosis of SVT of the upper limb, concluding that in the case of clinical suspicion, performing an ultrasound is necessary since D-dimer can be negative, even in 20% of SVT cases [52].

In VTE clinical management, there are some widely used and valid scores/models for predicting the risk of DVT and PE in the hospitalized medical patient (PADUA SCORE) [53] and the diagnostic probability of DVT–PE, such as the Geneva Score or the Wells score in outpatient populations [54,55].

A clinical score (ICARO SCORE) was proposed to evaluate SVT at risk of DVT, but the initially promising data [56] were not confirmed later [57]. Ultrasound scanning of the leg allows for the diagnosis of the SVT, evaluation of its extension, and the distance to the deep circulation, and excludes DVT [58,59,60]. The discussion surrounding the necessity of performing an ultrasound on a healthy leg and its economic viability is intriguing [61,62]. We believe that a complete ultrasound scan of both legs is advisable, as it allows for the acquisition of important information without excessive use of time.

Assessing the presence of CVD during an ultrasound scan is crucial to distinguish the onset of SVT in a varicose vein from a healthy vein. In this latter case, it potentially reveals an underlying neoplasm or other pathologies such as thrombophilia or autoimmune disease [36,63]. Furthermore, ultrasound findings are essential to evaluate the appropriate treatment of SVT (see Section 10).

## 9. Medical Treatment

Our analysis focused on the main trials that were divided by therapeutic options to determine the efficacy and safety of various medical treatments.

### 9.1. Graduated Elastic Compression Stockings

Boehler et al. analysed 80 patients in a randomised trial to determine the effectiveness of elastic compression (23–32 mmHg) alongside prophylactic heparin therapy and nonsteroidal anti-inflammatory drugs (NSAIDs). At 3 weeks, the addition of elastic stocking did not show significant benefits for the primary outcome (spontaneous and induced pain assessed during the 3 weeks) and secondary outcome (the number of analgesics consumed, skin erythema, thrombus length, D-dimer, and Quality of Life); however, after 1 week of elastic compression use, a reduction in thrombus extension was observed [64]. Although a consensus document recommended wearing elastic compression in addition to antithrombotic therapy [63], the most recent guidelines on the management of venous thrombosis [65] do not provide specific recommendations on the use of elastic compression in SVT. New studies are needed to determine the overall effectiveness and optimal duration of elastic compression therapy in SVT [66].

### 9.2. NSAIDs

The use of NSAIDs is common as a symptomatic treatment for SVT.

In the STENOX TRIAL [67], a double-blind trial involving 427 patients aged 18 and older with documented acute symptomatic SVT of the legs at least 5 cm long, participants were randomly assigned to receive different treatments once daily for 8 to 12 days. The treatments included subcutaneous enoxaparin sodium at a dose of 40 mg, subcutaneous enoxaparin at a dose of 1.5 mg/kg, oral tenoxicam, or placebo. The primary efficacy outcome was DVT occurrence, PE, or both between days 1 and 12. Among the secondary efficacy outcomes, DVT recurrence or extension toward the SFJ between days 1 and 12 was also evaluated. At day 12, NSAIDs significantly reduced the risk of SVT extension and/or recurrence by 54% compared to placebo (RR 0.46, 95% CI 0.27–0.78). Notably, differences in day 12 all-outcomes were not observed between the active treatment groups, except for a trend in favour of the low molecular weight heparins (LMWH) relative to the NSAIDs [67].

Additional research is needed to determine the role of NSAIDs in SVT therapy, whether alone or combined [68].

### 9.3. Unfractionated Heparins (UFH) and LMWH

Although anticoagulation is the primary function of heparins, other effects should also be taken into account, such as anti-inflammatory effects [69].

The first comparative trial that addressed the efficacy and safety of the various dosages of UFH was conducted in 2002 by Marchiori et al. who randomised 60 patients with SVT of the GSV. They compared unmonitored high doses of UFH (12,500 IU for one week followed by 10,000 IU) to prophylactic doses (5000 IU) for four weeks, and they showed that high doses were more effective in preventing VTE in such a setting (no patients vs. four patients treated with the prophylactic dose). Considering the total results after completing the planned six months of follow-up, the benefits of lower rates of symptomatic or asymptomatic events with the high doses compared to the prophylactic doses were maintained (3.3% vs. 20.0%, *p* = 0.05), without an increased risk of haemorrhagic complications [70].

In the aforementioned STENOX TRIAL [67], no significant benefit for DVT prevention for the 40 mg daily of enoxaparin or 1.5 mg/kg enoxaparin groups compared to placebo was found (0.9% and 1.0% vs. 3.6%, respectively); however, when assessing the combined incidence of DVT and SVT by day 12, there was a significant reduction in all active treatment groups compared to placebo. The incidence dropped from 30.6% in the placebo group to 8.3% and 6.9% in the 40 mg enoxaparin (*p* < 0.001) and 1.5 mg/kg enoxaparin (*p* < 0.001) groups, respectively. No deaths or major bleeding occurred during the study.

The Vesalio Investigators Group conducted a multicentre, prospective, controlled, and double-blind clinical trial involving patients with acute SVT of the GSV. For one month, patients were randomised to receive either high or low doses of LMWH. The primary objective of the trial was to compare the rate of asymptomatic and symptomatic extension of SVT and/or VTE. After a 3-month follow-up period, the cumulative rate of SVT progression and VTE complications did not show significant differences between patients receiving prophylactic doses (8.6%; 95% CI, 3.5–17.0) and therapeutic doses (7.2%; 95% CI, 2.8–15.1) of nadroparin. Of note, among patients receiving prophylactic doses of nadroparin who experienced complications, a higher proportion (71.4%) experienced thromboembolic events (all related to SVT progression) within the first month while still on treatment. In comparison, 33.3% of patients who received body-weight-adjusted full doses of nadroparin experienced complications (SVT progression in one patient). These observations suggested that the 1-month therapeutic scheme of the trial may offer more effective protection against the progression of thrombophlebitis. However, this effect appears to diminish after discontinuation of the drug [71].

The STEFLUX trial [72] enrolled 664 patients with isolated SVT of the lower limbs, with a minimum length of at least 4 cm. Exclusion criteria included SVT in the saphenous or short saphenous vein within 3 cm of the SFJ or sapheno–popliteal junction (SPJ), respectively. Patients were randomly assigned in a double-blind fashion to receive the following: (1) intermediate dose of LMWH (Parnaparin 8500 IU once daily) for 10 days and placebo for 20 days; (2) intermediate dose of LMWH for 30 days (Parnaparin 8500 IU once daily for 10 days followed by 6400 IU once daily for 20 days); or (3) prophylactic dose of LMWH (Parnaparin 4250 IU once daily) for 30 days.

The primary outcome (a composite of symptomatic and asymptomatic DVT, PE, and SVT recurrence) was significantly lower with the use of a 30-day intermediate dose (1.8%) compared to the shorter intermediate LMWH dose (15.6%) and the prophylactic one (7.3%). Therefore, data from the STEFLUX study indicated that an intermediate dose of LMWH for 30 days is more effective than a shorter duration or prophylactic dose. However, after 3 months, the benefit of the intermediate dose treatment versus the prophylactic one disappeared with a “catch-up” of the curves (*p* = 0.06). This nuanced approach identifies the heterogeneity in patient responses and underscores the importance of tailoring the duration and intensity of LMWH treatment based on individual clinical profiles in the context of lower limb SVT.

A prospective study by Nikolakopoulos et al. enrolled 147 patients with isolated SVT (length > 5 cm and distance > 3 cm from the SFJ). Patients were divided into 2 groups, the first receiving Tinzaparin at variable dosages for 60 days (group A), and the second (group B) receiving an intermediate dose of Tinzaparin (75.0% of the therapeutic) for 90 days. The recurrence of thromboembolic events was statistically significant in the shorter therapy group (*p* = 0.004). Although the study had several limitations (non-randomised trial and absence of standardization of the Tinzaparin dosage), according to the results, the extended three-month regimen of Tinzaparin could be a valuable option to prevent DVT complications in specific patients with SVT, such as superficial axial vein thrombosis and multiple thrombosed superficial sites [73].

Recently, a pooled analysis of two trials with Tinzaparin treatment at an intermediate dosage (131 IU/kg) was performed. Among 956 patients, Tinzaparin administered at an intermediate dose for 30 days resulted in an effective and safe treatment for SVT. Interestingly, the duration of treatment was not associated with recurrent thromboembolic events (*p* = 0.46); however, the length of the thrombus at the index event was significantly associated with a higher risk of recurrence of VTE. Future studies should identify subgroups of patients who are likely to benefit from longer durations of treatment [74].

### 9.4. Fondaparinux

Fondaparinux is a synthetic pentasaccharide consisting of the minimum sequence of heparin that interacts with AT and is a selective inhibitor of factor Xa. Its mechanism of action on increased lysis of thrombus seems to be dual: the change of the structure of the clot and the inactivation of thrombin-activable fibrinolysis inhibitor [75].

The CALISTO TRIAL, a randomised, double-blind study, investigated the efficacy and safety of fondaparinux in treating acute SVT of the legs. Among the 3002 enrolled patients with confirmed SVT of at least 5 cm, those receiving fondaparinux (2.5 mg once daily) for 45 days demonstrated a reduction in the outcomes compared to the placebo group. Indeed, the primary efficacy outcome (a composite of death, symptomatic PE, DVT, symptomatic extension to the SFJ, or symptomatic SVT recurrence at day 47) occurred in 0.9% of the fondaparinux group versus 5.9% in the placebo group, with 85% relative risk reduction (95% CI, 74 to 92; *p* < 0.001). The incidence of PE or DVT was significantly lower in the fondaparinux group (0.2% vs. 1.3%; 95% CI, 50 to 95; *p* < 0.001), with consistent risk reductions at day 77. Major bleeding was rare (one patient in each group), and serious adverse events were reported in 0.7% of patients with fondaparinux and 1.1% with placebo. In summary, the treatment with fondaparinux at a 2.5 mg once-daily dose for 45 days demonstrated efficacy and safety for the treatment of acute SVT [76].

### 9.5. Direct Oral Anticoagulants (DOACs)

DOACs have played a crucial role in stroke prevention in atrial fibrillation and VTE treatment for several years [77,78]. However, there is a limited body of evidence on their effectiveness in treating SVT.

The phase 3b trial SURPRISE, including 472 SVTs with at least 5 cm extension and at least one additional risk factor (older than 65 years, male sex, previous VTE, cancer, autoimmune disease, or thrombosis of non-varicose veins), found rivaroxaban (10 mg) to not be inferior to fondaparinux (2.5 mg) over a 45-day span, showing comparable outcomes in terms of DVT/PE, SVT progression/recurrence, and all-cause mortality. Notably, neither group experienced major bleeding, suggesting that rivaroxaban, being less expensive, could be a viable alternative to fondaparinux in high-risk SVT patients [79].

Another study attempted to compare a 10 mg dose of rivaroxaban against a placebo for 45 days in the treatment of SVT of at least 5 cm diagnosed with or without ultrasound; however, this study failed to recruit enough patients to achieve statistical significance. Although not statistically relevant, the data that emerged showed that fewer patients in the rivaroxaban group were using analgesics and anti-inflammatory agents on day 7 [80].

## 10. Practical Approach in the Patient with SVT

According to guidelines [78], if the superficial thrombus extends to deep circulation or the distance from the SFJ/SPJ is less than 3 cm, the recommended treatment is a full-dose anticoagulant. Conversely, if SVT has an extension >5 cm but is quite far from the femoral or popliteal vein (>3 cm), the recommended therapy is fondaparinux 2.5 mg daily for 45 days or rivaroxaban 10 mg. If the SVT is less than 5 cm in extent and is located quite far from deep circulation, NSAIDs with elastic stockings can be considered [78]; however, serial ultrasound scans are advisable to evaluate thrombus progression or resolution.

An interesting observation about the cost-effectiveness of treatment with fondaparinux came from Blondon and coworkers, who evaluated in 10,000 patients the costs using a decision-tree analysis. The authors analysed clinical events, quality-adjusted life-years, costs, and incremental cost-effectiveness ratios using a lifetime time horizon from a healthcare system perspective [81]. The data suggested that treatment with fondaparinux for 45 days was not effective in the case of isolated SVT of the legs. The researchers postulated that a money-valuable strategy could be to treat patients with a higher short-term risk of VTE and decrease the duration of treatment.

Figure 1 shows our suggested approach.

## 11. Discussion and Future Directions

The frequent underestimation and the misconception about the hazard of SVT in general practice probably prevented epidemiological studies [7,8,9,10,11,12,13,14] from accurately identifying the incidence and prevalence of SVT in the general population. As previously discussed, SVT should not be considered a benign disease. Indeed, the coexistence of DVT and PE at SVT diagnosis has been reported at rates as high as 18% and 6.3%, respectively, according to the meta-analysis discussed above [40].

To assess the extent and risk of an individual patient, ultrasound is essential, as the therapeutic strategy changes according to the extension of SVT. As it is a non-invasive technique, it can also be routinely used for serial evaluation to document any progress or resolution of the disease, particularly when anticoagulant therapy is not administered to the patient.

Additional studies are necessary to provide additional elements that aid clinicians in stratifying risk by choosing the most sustainable approach and preventing long-term complications.

Open questions remain in the management of SVT, mainly whether all SVT cases near the deep circulation have to be treated with the full therapeutic anticoagulant dose for three months, if the recommended treatment with 2.5 mg of fondaparinux is sufficient for long-extended SVT, and what is the optimal treatment after the suggested 45 days of treatment.

Patients with SVT involving SFJ were excluded from SVT treatment trials, and experts suggest using an anticoagulant dose for 3 months, similar to what is done for DVT [82].

An analysis of RIETE registry data which compared the use of therapeutic doses (227 patients) with the prophylactic ones (147 patients) for the treatment of SVT extended to the junction, showed that the use of subtherapeutic doses of anticoagulants was not less effective as the therapeutic one for the management of such patients (1.3% vs. 2.7%, *p* = 0.56). Although non-randomised, these data pose additional uncertainties about optimal therapy [83].

Recently, during a survey, heterogeneity was found in the overall management of patients with SVT and in patients with extensive forms of thrombosis. Although most clinicians considered SVT next to the femoral vein segment similar to DVT (89%), only 57% prescribed full anticoagulation for three months [84].

In patients with isolated SVT, the INSIGHTS-SVT study recorded prospective information on the 3-month evaluation of symptomatic DVT, PE, and progression or recurrence of SVT. Out of the 1150 patients included, 4.7% had recurrent or extended SVT, 1.7% had DVT, and 0.8% had PE [85]. After adjusting for propensity score and treatment duration, it was observed that fondaparinux had a lower primary outcome than LMWH. The multivariable analysis showed that factors associated with the primary outcome were a previous SVT event (HR 2.3, 95% CI 1.44–7.78, *p* = 0.001), age per year (HR 0.97, 95% CI 0.96–0.99, *p* = 0.008), duration of treatment per week (HR 0.92, 95% CI 0.83–0.99, *p* = 0.046), and thrombus length (HR 1.03, 95% CI 1.02–1.05, *p* < 0.001). Fondaparinux treatment was taken for at least four weeks by a majority of patients (70%, 487 out of 696), while LMWH treatment was taken by only 28% (85/305) patients. The reasons for a shorter duration of LMWH therapy were unclear. Compared to LMWH, patients treated with fondaparinux experienced fewer recurrent VTE events if treatment was extended for more than 38 days (3.7 vs. 10.5%; HR 0.37, 95% CI 1.03–7.03; *p* = 0.044) [85].

An extension of the INSIGHT-SVT study with up to 12 months of follow-up confirmed the increased risk of VTE and, therefore, a longer therapeutic approach could be reasonable in patients with risk factors [86]. Another sub-analysis of the INSIGHTS-SVT register revealed that cancer patients were at risk of thromboembolic complications despite antithrombotic treatment. While most events occurred within 3 months (HR: 3.63, 95% CI 1.79–7.35, *p* < 0.001), the risk remained high with up to 1 year of follow-up (HR: 2.40, 95% CI 1.30–4.45, *p* = 0.005). These data emphasize the importance of considering the high risk of VTE for cancer patients and considering longer and more intensive anticoagulation on an individual basis [87].

The TROLL registry assessed the cumulative risk of VTE in 229 patients with isolated SVT treated for 45 days, with a 1-year prevalence of 4.6% and a 5-year prevalence of 15.9%. Even though anticoagulant treatment was used, the risk of VTE after isolated SVT at high risk was significant, whereas bleeding complications were minimal [88]. In addition, a recent meta-analysis revealed that longer anticoagulant treatment is linked to decreased VTE rates in patients with SVT of lower limbs. The duration of anticoagulant treatment for 14 days was associated with the highest VTE rate. Future studies should determine the most effective duration of anticoagulation treatment [89].

A post-hoc analysis of STENOX data has revealed four key features that could potentially predict DVT or PE three months after isolated SVT. They include male sex (OR: 2.05, 95% CI 1.23–3.40, *p* = 0.006), a history of VTE (OR: 1.98, 95% CI 1.05–3.76, *p* = 0.03), severe chronic venous insufficiency (OR: 2.50, 95% CI 1.05–5.93, *p* = 0.04), and a short interval between the onset of symptoms and the diagnosis (OR: 2.65, 95% CI 1.30–5.42, *p* = 0.007). However, when evaluating long-term outcomes, it is important to take into account the study’s relatively short duration of therapy [90].

A post hoc analysis of the STEFLUX study showed that some factors, such as the family history of VTE (OR: 2.6, 95% CI 1.4–4.8, *p* < 0.001) and the absence of varicose veins (OR: 2.6, 95% CI 1.3–5.0, *p* = 0.007), conferred higher risk for the development of VTE, especially after the suspension of treatment. Again, these observations underline the need to identify the subjects in need of individualized therapy in terms of duration and dosage [91].

As mentioned above, there is much uncertainty about the continuation of therapy after the recommended 45 days. An interesting ongoing study, the METRO STUDY, enrolled 650 patients to evaluate the effectiveness of 50 mg of mesoglycan twice a day versus a placebo after 45 days of therapy with 2.5 mg of fondaparinux daily in SVTs of at least 5 cm and far more than 3 cm from the SFJ. The primary efficacy outcome of the study will be an objectively documented symptomatic recurrence or extension of SVT, symptomatic or asymptomatic DVT (proximal or distal), symptomatic PE, or death [92].

Finally, new drugs, such as FXI inhibitors, are under study, and the first set of data emerging from the meta-analysis confirmed their effectiveness and safety in some thrombotic clinical settings [93]. Starting specific and dedicated studies to treat SVT would be very useful.

## 12. Conclusions

SVT is not a benign disease since it is not free from complications, including extension to the deep venous system and embolization at the level of the pulmonary vascular bed. A prompt clinical and ultrasound evaluation allows for the set up of an appropriate therapy that, in the case of SVT with an extension greater than 5 cm and adequately distant from the deep circle, consists of a six-week treatment with 2.5 mg of fondaparinux mg daily. However, in certain cases and clinical contexts, treatment of different durations and dosages may be considered, but further studies are needed to assess the real benefit and economic impact.

## Figures and Tables

**Figure 1 healthcare-12-00500-f001:**
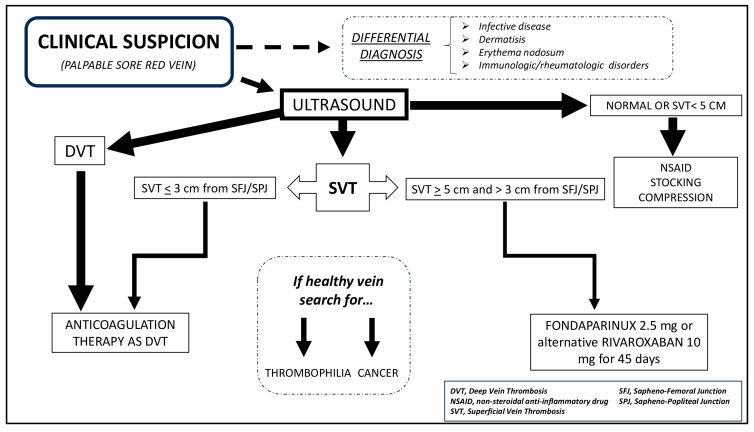
Practical approach in the patient with SVT.

**Table 1 healthcare-12-00500-t001:** SVT risk factors and prevalence of genetic thrombophilia.

Trial	Patients	Risk Factor	Prevalence	ODDS Ratio
Karathanos et al. [23]	128 SVT+ varicose veins vs. 102 varicose veins	Protein S deficiency		OR 5.07 (95% CI 1.32–19.42), *p* = 0.018
		Obesity		OR 5.50 (95% CI 2.16–13.99), *p* = 0.001
		Male patients		OR 2.02 (95% CI 1.01–4.02), *p* = 0.047
		Age > 70 years		OR 4.11 (95% CI 2.06–8.18), *p* = 0.001
		Age > 45 years		OR 2.28 (95% CI 1.29–4.01)
De Moerlose et al. [24]	112 SVT vs. 180 healthy controls	Factor V Leiden	14.3%	OR 2.51 (95% CI 1.04–6.24)
		Factor II G20210A	3.6%	OR 3.28 (95% CI 0.46–36.84)
Margaglione et al. [25]	105 SVT	Factor V Leiden	16.2%	
		Factor II G20210A	7.6%	
Martinelli et al. [26]	63 SVT vs. 537 healthy controls	Factor V Leiden	15.9%	OR 6.1 (95% CI 2.6–14.2)
		Factor II G20210A	9.6%	OR 4.3 (95% CI 1.5–12.6)
		AT III, protein C, and protein S deficiency	10.2%	OR 6.64 (95% CI 2.6–46.2)
Lucchi et al. [27]	73 SVT non-varicose veins	Factor V Leiden	51%	
		MTHFR mutation	38.3%	
		Factor II G20210A	3.3%	
		Protein C deficiency	1.6%	
		Protein S deficiency	1.6%	
		Antiphospholipid antibodies	8.3%	
Legnani et al. [28]	1294 SVT vs. 1294 healthy controls	Protein C and protein S deficiency		OR 12.2 (95% CI 2.87–52.1), *p* < 0.001
		Factor V Leiden		OR 2.97 (95% CI 2.15–4.10), *p* < 0.001
		Factor II G20210A		OR 1.37 (95% CI 0.95–1.98), *p* = 0.090
		Antiphospholipid antibodies		OR 2.92 (95% CI 0.91–9.40), *p* = 0.073
		Combined alterations		OR 26.1 (95% CI 3.48–196.3), *p* = 0.002
		High factor VIII Levels		OR 0.96 (95% CI 0.65–1.43), *p* = 0.862

Legend: OR: Odds Ratio and CI: Confidence Interval.

**Table 2 healthcare-12-00500-t002:** Risk and prevalence of DVT/PE at SVT diagnosis.

Trial	Patients	DVT/PE
STEPH [7]	171	24.6% DVT 4.7% PE
POST [38]	844	23.5% DVT 3.9% PE
OPTIMEV [39]	788	22.5% DVT 6.8% PE
Di Minno MN et al. [40]	4300 22 studies	18% DVT 6.4% PE *

Legend: DVT: Deep Venous Thrombosis; PE: Pulmonary Embolism. * in asymptomatic.

**Table 3 healthcare-12-00500-t003:** Risk factors for DVT/PE in SVT.

Obesity	Age > 75 years, male sex
Trauma	Active cancer
Hypercoagulable states	Inpatient status
Use of oral contraceptives or hormonal therapies	Antithrombin, protein C, and protein S deficiency
Previous DVT/PE	Factor V Leiden
Infusion of endothelium-damaging substances	Autoimmune diseases
Pregnancy	SVT on non-varicose veins

## Data Availability

Data sharing not applicable.

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
