# Peer review of "Superficial Venous Thrombosis: A Comprehensive Review"

_healthcare, 2024, doi:10.3390/healthcare12040500_

Round 1

Reviewer 1 Report

Comments and Suggestions for Authors

This is a comprehensive review of the literature for diagnosis and management of SVT.  I would have preferred more summation and analysis than having to read all the study results  individually.

Intro:  Virchow’s triad mainly applies to venous thrombosis

Predisposing factors:  I believe the term prevalence is used inappropriately here- Relative risk, odds ratio, should be used to describe risk factors in cohort or case control studies.  Would be helpful to see a table of risk factors ranked by frequency.

Epidemiology:  How were these studies done?  National health statistics?  Large databases?  Also, how was SVT diagnosed- clinical symptoms or screening US?

Complications:  How were PEs and DVTs screened and  diagnosed?  Were the DVTs extension of the SVTs or in different location?  Were the PEs subsegmental?  It’s no surprise that SVTs would cause PEs, but likely too small to cause clinical consequences.  What is the explanation for divergent results among all these studies?  Likely the study design and population studies were very different.

Treatment:  this section could be condensed and summarized.  Again, wish to learn how these outcomes were measured: symptoms? Screening US?

I believe it’s important to communicate that while the efficacy of different treatments may be statistically significant, the absolute risk of complications is still very low and patients may wish to forgo treatment entirely.

Practical approach:  What the practicing hematologist needs to know are the odds of extension, embolization, clinical symptoms in order to determine when treatment should be recommended.  Serial US is considered when anti-coagulation is not given, but that wasn’t mentioned here.

Comments on the Quality of English Language

There were some strange word choices:

ex/

Abstract ln 10:  recognizes?

Epidemiology:  ln 88:  scenery?

Author Response

This is a comprehensive review of the literature for diagnosis and management of SVT. I would have preferred more summation and analysis than having to read all the study results individually.

R: We are sorry, we made several changes hoping to improve the work and its readability

Intro: Virchow’s triad mainly applies to venous thrombosis

R: We concur with the reviewer, we specified in the text

Predisposing factors: I believe the term prevalence is used inappropriately here- Relative risk, odds ratio, should be used to describe risk factors in cohort or case control studies. Would be helpful to see a table of risk factors ranked by frequency.

R: We corrected the title of the paragraph, provided (when available) hazard ratio and odds ratio, and improved the table by adding data on thrombophilic factors and risk-related factors.

Epidemiology: How were these studies done? National health statistics? Large databases? Also, how was SVT diagnosed- clinical symptoms or screening US?

R: We have improved the description according to the available data.

Complications: How were PEs and DVTs screened and diagnosed? Were the DVTs extension of the SVTs or in different location? Were the PEs subsegmental? It’s no surprise that SVTs would cause PEs, but likely too small to cause clinical consequences. What is the explanation for divergent results among all these studies? Likely the study design and population studies were very different.

R: As stated in the text, studies are very heterogeneous, most of them are old, and the diagnosis of SVT has been often clinical, therefore, it is complicated to draw any conclusions. We discussed such issues in the text, and we reported on the current available evidence.

Treatment: this section could be condensed and summarized. Again, wish to learn how these outcomes were measured: symptoms? Screening US?

R: We also changed this section according to the comments of other reviewers.

I believe it’s important to communicate that while the efficacy of different treatments may be statistically significant, the absolute risk of complications is still very low and patients may wish to forgo treatment entirely.

R: We understand the auditor’s point of view by agreeing that the risk is very low in some cases, indeed we discussed it in terms of economic sustainability; however, we cited several studies in which residual risk was reported despite treatment. The occurrence of a thrombotic event, especially if it is a pulmonary embolism, is accompanied by a series of important sequelae for the patient.

Practical approach: What the practicing hematologist needs to know are the odds of extension, embolization, clinical symptoms in order to determine when treatment should be recommended. Serial US is considered when anti- coagulation is not given, but that wasn’t mentioned here.

R: We clarified the odds and specific risk factors for complications during the discussion. Additionally, we discussed the possibility of conducting multiple ultrasound scans in a particular clinical setting.

Comments on the Quality of English Language There were some strange word choices:
ex/
Abstract ln 10: recognizes?

R: We apologize, we have corrected it in the text.

Epidemiology: ln 88: scenery?

R: We apologize, we have corrected it in the text.

Reviewer 2 Report

Comments and Suggestions for Authors

Dear Authors! Thank you for the interesting and useful review that you conducted. 

1. Lines 25-37 have to be deleted either shortened as this data are not related to SVT but to uncomplicated varicose veins and deep 

2. Line 110. SVT but not STV

3. Line 117. It is clearly meant not CVI classification, but CVD (chronic venous disease) classification.

4. Table 1 - there is no need to use SVT and PT abbreviations in a column 2. 

5. Line 232. Word "complications" should be deleted.

6. Line 256. It's obviously that 33.3% is not a low percentage. Please, put this data on a proper place in the statement.

7. It would be logical to make a subsection on diagnostics of SVT somewhere in the beginning of the paper. I recommend  to use part of the data from subsection 9 to create diagnostics subsection.

Comments on the Quality of English Language

I believe a native speaker should read paper and correct some lexics

Author Response

Dear Authors! Thank you for the interesting and useful review that you conducted.

R: we thank the reviewer for his valuable comment, we are glad that our work was found interesting

1. Lines 25-37 have to be deleted either shortened as this data are not related to SVT but to uncomplicated varicose veins and deep

R: We shortened that period focusing only on the first description of varicose vein and thrombosis (line 25-32)

2. Line 110. SVT but not STV

R: We apologize, we have corrected it in the text.

3. Line 117. It is clearly meant not CVI classification, but CVD (chronic venous disease) classification.

R: We apologize, we have corrected it in the text.

4. Table 1 - there is no need to use SVT and PT abbreviations in a column 2.

R: We have corrected it, as suggested by the reviewer.

5. Line 232. Word "complications" should be deleted.

R: We apologize, we have corrected it in the text.

6. Line 256. It's obviously that 33.3% is not a low percentage. Please, put this data on a proper place in the statement.

R: in the text we referred to the fact that the percentage of events in this group was lower than the patients receiving prophylactic dose. Anyway, we removed "lower" leaving only the percentage to avoid confusion.

7. It would be logical to make a subsection on diagnostics of SVT somewhere in the beginning of the paper. I recommend to use part of the data from subsection 9 to create diagnostics subsection.

R: We thank the reviewer for his suggestion, we modified the manuscript adding the paragraph “8. Diagnosis of SVT”.

I believe a native speaker should read paper and correct some lexics

R: we corrected the text as suggested

Reviewer 3 Report

Comments and Suggestions for Authors

This is an extensive and really comprehensive review on superficial vein thrombosis. The paper is very easy to read and fully comprehensible.

I have only minor issues and few typos to report:

1. page 3, line 110: "cost related to SVT"

2. page 4, Table 1: at the bottom of the table, citing the work of Di Minno, typo "in asymptomatic"

3. page 6, line 262:  "The Steflux trial enrolled 664 patients", the word "of must be removed

4. page 6-page 7, lines 270-277: when reporting data of the STEFLUX trial, Authors must add a statement that at the 3-months follow-up the benefit of the intermediate doses treatment vs the prophylactic dosese treatment disappear and there is the catch-up phenomenon of the curves, showing that a 30-day treatment course is not sufficient for SVT patients

5. page 8, line328: "anti-inflation" must be replaced by "anti-inflammatory"

6. page 8, line 331: "tender palpable cord on the skin"

7. page 8, line 342:  ".....such as the Geneva score or the Wells score in the out patients"

8. page 8, line 347: "Interesting is the debate...."

9. page 8 , lines 348-350: please rephrase the full statement, it is non clear what the Authors mean

10. page 10, line 414: "An extension of the INSIGHTS-SVT study....."

Author Response

This is an extensive and really comprehensive review on superficial vein thrombosis. The paper is very easy to read and fully comprehensible.

R: we thank the reviewer for his valuable comment, we are glad that our work was found interesting

I have only minor issues and few typos to report:
1.page 3, line 110: "cost related to SVT"
R: We apologize, we have corrected it in the text
2. page 4, Table 1: at the bottom of the table, citing the work of Di Minno, typo "in asymptomatic" R: we have corrected it in the text

3. page 6, line 262: "The Steflux trial enrolled 664 patients", the word "of must be removed

R: We removed it

4. page 6-page 7, lines 270-277: when reporting data of the STEFLUX trial, Authors must add a statement that at the 3- months follow-up the benefit of the intermediate doses treatment vs the prophylactic dosese treatment disappear and there is the catch-up phenomenon of the curves, showing that a 30-day treatment course is not sufficient for SVT patients

R: We concur with the reviewer, we added in the text “However, after 3 months the benefit of the intermediate dose treatment vs the prophylactic one disappeared with “catch-up” of the curves (p=0.06)”.

5. page 8, line328: "anti-inflation" must be replaced by "anti-inflammatory"

R: we have corrected it in the text

6. page 8, line 331: "tender palpable cord on the skin"

R: we have corrected it in the text

7. page 8, line 342: ".....such as the Geneva score or the Wells score in the out patients"

R: we have corrected it in the text

8. page 8, line 347: "Interesting is the debate...."

R: we have corrected it in the text

9. page 8 , lines 348-350: please rephrase the full statement, it is non clear what the Authors mean

R: We concur with the reviewer, the sentence was unclear, we have corrected the statement as the following: “We believe that a complete ultrasound scan of both legs is advisable as it allows to acquire important information without excessive use of time”.

10. page 10, line 414: "An extension of the INSIGHTS-SVT study....."

R: we have corrected it in the text

Round 2

Reviewer 1 Report

Comments and Suggestions for Authors

I appreciate the authors' responses to my comments.  The manuscript is much improved.  I have only a few minor semantic suggestions:

P 1 ln 10 "identifies several underlying causes" change to "may have several different underlying causes"

p1 ln 15 Mandatory is too strong:  recommended or essential better

pg 1 ln 16 SVT may be costly

pg 1 ln 18 analyse should be review

p 1 ln 38 troubling should be possible

p 1 ln 39 analyse should be review

p 1 ln 44 incidence is estimated to be

p 2 ln 45 dvt, which is estimated to be ...

lp 2 ln 52 data should be true prevalence

p 2 ln 94 is....causative factor should be are... possible risk factors

p 4 ln 126 government should be health care

p 4 ln 129 disease should be complication or condition

p 4 ln 140 was should be may have been

p 4 ln 142 - what is crescent?

p 5 ln 185 - Previously it is mentioned that chronic venous insufficiency is caused by SVT, but that is not mentioned in possible complications.

p 9 ln 392 - Is this the authors recommendations or comes from other guidelines?  Should be clear.

p 9 ln 294 < and less than is redundant- remove one

p 9 ln 395 > 5 cm but

p 9 ln 396 - product specificity is unfortunate, LMWH or DOACs is preferred.  Can mention whether preventive dosing or treatment dosing is recommended

p 12 ln 497 :  interesting could be changed to very useful

p 12 ln 501:  I don't think the deep venous system or pulmonary circulation are circles -maybe switch to deep venous circulation and pulmonary vascular bed

p 12 ln 502 clinical and ultrasound evaluation 

Comments on the Quality of English Language

Still with some funny word choices, attempted to point out in my review

Author Response

Dear Auditor, we thank you once again for your contribution in improving our work. We have corrected all the grammatical errors as you suggested, for the sake of brevity we summarize some crucial points:

p 5 ln 185 - Previously it is mentioned that chronic venous insufficiency is caused by SVT, but that is not mentioned in possible complications.

R: We agree that superficial venous thrombosis predisposes to chronic venous disease, however, we have not found any important data to support the association between chronic venous disease secondary to superficial venous thrombosis and complications such as deep vein thrombosis and pulmonary embolism. If the reviewer is aware of this literature, we are grateful for his contribution to improving the work.

p 9 ln 392 - Is this the authors recommendations or comes from other guidelines?  Should be clear.

R: We clarified it in the text.

p 9 ln 396 - product specificity is unfortunate, LMWH or DOACs is preferred.  Can mention whether preventive dosing or treatment dosing is recommended

R: We have reported these data because in the guidelines we have taken as a reference (ref 78) in points 19 and 20 it refers specifically to Fondaparinux 2.5 mg and Rivaroxaban 10 mg.